# Preparation of MgO Self-Epitaxial Films for YBCO High-Temperature Coated Conductors

**DOI:** 10.3390/mi14101914

**Published:** 2023-10-08

**Authors:** Fei Yu, Yan Xue, Chaowei Zhong, Jiayi Song, Qiong Nie, Xin Hou, Baolei Wang

**Affiliations:** 1School of Electronic and Communication Engineering, Shenzhen Polytechnic, Shenzhen 518000, China; yufei198275@szpt.edu.cn (F.Y.); zhongchaowei2023@163.com (C.Z.); songjiayi0220@163.com (J.S.); 2Ningbo Weierskeller Intelligent Technology Co., Ltd., Ningbo 315210, China; wangbaolei2023@163.com; 3Materials Science and Engineering, Tianjin University, Tianjin 300072, China; houxin@tju.edu.cn; 4School of Foreign Languages and Trade, Guangzhou City Construction College, Guangzhou 510006, China

**Keywords:** ion beam-assisted deposition, magnesium oxide, homogeneous epitaxy, fluorene

## Abstract

Ion beam-assisted deposition (IBAD) has been proposed as a promising texturing technology that uses the film epitaxy method to obtain biaxial texture on a non-textured metal or compound substrate. Magnesium oxide (MgO) is the most well explored texturing material. In order to obtain the optimal biaxial texture, the actual thickness of the IBAD-MgO film must be controlled within 12nm. Due to the bombardment of ion beams, IBAD-MgO has large lattice deformation, poor texture, and many defects in the films. In this work, the solution deposition planarization (SDP) method was used to deposit oxide amorphous Y_2_O_3_ films on the surface of Hastelloy C276 tapes instead of the electrochemical polishing, sputtering-Al_2_O_3_ and sputtering-Y_2_O_3_ in the commercialized buffer layer. An additional homogeneous epitaxy MgO (epi-MgO) layer, which was used to improve the biaxial texture in the IBAD-MgO layer, was deposited on the IBAD-MgO layer by electron-beam evaporation. The effects of growth temperature, film thickness, deposition rate, and oxygen pressure on the texture and morphology of the epi-MgO film were systematically studied. The best full width at half maximum (FWHM) values were 2.2° for the out-of-plane texture and 4.8° for the in-plane texture for epi-MgO films, respectively. Subsequently, the LaMnO_3_ cap layer and YBa_2_Cu_3_O_7_-x (YBCO) functional layer were deposited on the epi-MgO layer to test the quality of the MgO layer. Finally, the critical current density of the YBCO films was 6 MA/cm^2^ (77 K, 500 nm, self-field), indicating that this research provides a high-quality MgO substrate for the YBCO layer.

## 1. Introduction

REBa_2_Cu_3_O_7−_x (REBCO) superconducting conductors exhibit excellent electrical properties and broad application prospects in the fields of transmission cables, strong magnets, motors, and current limiters [1,2,3,4,5,6]. In order to realize the application of high-temperature superconducting materials in the field of transmission cables, the critical current (*Ic*) must be as high as 200 A/cm at 77 K self-field [7]; therefore, the REBCO conductors must be epitaxially grown on the metal substrates with biaxial texture by the thin film deposition method to avoid grain boundary misalignment [8]. However, the direct deposition of REBCO films on metal substrates causes the diffusion of metal elements from the metal substrate to the superconducting layer [9]. In addition, the oxygen element is easy to diffuse to the metal substrate, thereby reducing the mechanical properties of the metal substrate [10]. Therefore, an oxide buffer layer structure must be introduced between the metal substrate and the superconducting layer [11,12,13,14].

The functions of the buffer layers can be described as follows. First, the mutual diffusion of elements is blocked between the metal substrate and the REBCO layer. The flat and dense oxide buffer films can prevent direct contact between the superconducting layer and the REBCO layer, effectively blocking the mutual diffusion of elements [15,16,17]. Second, the adhesion of YBCO layer deposited on oxide buffer layers is much better than that on the metal substrates [18,19,20]. Finally, the biaxial texture is induced in the REBCO layer by the epitaxial grown on a textured oxide buffer layers [21,22,23].

At present, there are three well-explored technologies for obtaining biaxial texture, i.e., rolling-assisted biaxial texture (RABiTS), inclined substrate deposition (ISD), and ion beam-assisted deposition (IBAD) [24]. The biaxial texture can be obtained by the thin film epitaxy method on a non-textured metal or compound substrate in the IBAD technology [25]. The interaction between ions and deposited atoms promotes grain rearrangement in the buffer layers [26]. There are some critical issues that need to be addressed in the preparation process of buffer layers, including obtaining high-quality biaxial texture [27,28,29], reducing the production costs [30], and improving the uniformity of the buffer layer [31].

Yttria Stabilized Zirconia (YSZ) (ZrO_2_–Y_2_O_3_ stabilized) is one of the earliest materials to be used as a biaxially textured template for REBCO. YSZ is cubic and has a lattice constant of 5.14 A°. In 1992, Iijima et al. reported that a sharp biaxially aligned structure of YSZ films grown on alloy as a substrate could be developed by IBAD technology [32]. The YSZ films could be successfully used for REBCO films and resulted in a demonstration of high Jc values [33]. The in-plane FWHM of the XRD in-plane scan ranges from 7–15° [34,35,36]. The most significant disadvantage in using YSZ as a template material is that the texture evolution occurs rather slowly, requiring about 1 µm film thickness to achieve the optimum in-plane texture [35]. This requires a deposition time of several hours to achieve the desired thickness of the template material. This has limited YSZ as a texturing material in many applications. 

The biaxial texture formation of magnesium oxide (MgO) grown by IBAD was first reported by the researchers from Stanford University [37]. In contrast to IBAD-YSZ, IBAD-grown MgO has been observed to have optimum biaxial texture at about 10 nm of film thickness, while IBAD-YSZ requires typically about 700–1000 nm thickness of film. From a practical point of view, IBAD-MgO growth is about 100 times faster than that for IBAD-YSZ, which translates to significantly lower costs. The current work focuses on the production of long lengths of IBAD-MgO. The textured MgO template has been adopted world-wide as the technology of choice from the USA to Japan to Korea [11,15,32,36]. However, improvements in the quality of IBAD-MgO have seemingly stalled. The standard coated-conductor buffer structure using IBAD-MgO is composed of a LaMnO_3_ (LMO)/epitaxial-MgO/IBAD-MgO/ amorphous-Y_2_O_3_/amorphous-Al_2_O_3_ five-layer stack. In our previous study, the solution deposition planarization (SDP) method, which reduced the preparation cost and process complexity of the REBCO superconducting strip, was used to deposit oxide amorphous films on the surface of Hastelloy C276 tapes [38]. The SDP-Y_2_O_3_ layer can replace the three processes, including electrochemical polishing, sputtering-Al_2_O_3_, and sputtering-Y_2_O_3_ in the commercialized buffer layer. 

In our previous research, LaMnO_3_ (LMO)/IBAD-MgO/SDP-Y_2_O_3_ has been used to sever the buffer template for YBCO films [38]. However, the buffer layer shows poor biaxial texture with an out-of-plane full width at half maximum (FWHM) value of 3.5° (the rocking curve of LMO (002) peak) and in-plane FWHM value of 7.2° (the *Φ*-scan of LMO (222) peak) in the buffer layer, respectively. It should be noted that sufficient critical currents can hardly be obtained with the in-plane texture over 5° [39]. Therefore, it is necessary to reduce the biaxial texture. Y. Yamada et al. have deposited CeO_2_ on IBAD-MgO to improve the biaxial texture. However, the best in-plane FWHM value is 6° in their research [40].

In this paper, we prepared epi-MgO films using electron beam evaporation and systematically investigated the influence of the growth temperature, film thickness, deposition rate, and oxygen pressure on the texture and morphology of the MgO films. It is necessary to obtain the MgO film with the best biaxial texture, which would provide an excellent growth template for YBCO superconducting thin films. Finally, the LMO cap layer and YBCO superconductor films are fabricated on the MgO films to testify the function of the epi-MgO films.

## 2. Experiment

In this work, a home-made electron beam evaporation, which offers the ability to evaporate metal oxide materials with a high melting point, was used to deposit epi-MgO films on 10 nm-thick IBAD-MgO films. In this experiment, the vacuum chamber was pumped down to 10^−5^ Pa using a molecular pump. Subsequently, the growth temperature (150~500 °C) required for the experiment was controlled by a home-made heating device. A reel-to-reel system enabled the continuous preparation of long tapes for the dynamic deposition of the epi-MgO films. When the metal oxide films were prepared using electron beam evaporation, the ionic bonds of the materials could be easily disrupted by the electron beam, potentially causing the composition of the film material to deviate and affecting the microstructure of the film. Therefore, it was necessary to introduce oxygen to maintain the proper proportion of MgO film composition during the epi-MgO deposition process. Oxygen was introduced to the evaporation chamber through an oxygen valve. The quartz crystal microbalance (QCM) rate monitor and ion probe were used to measure the electron beam evaporation rate and the ion beam current density, respectively. The QCM was installed at the location of the substrate. Subsequently, the LMO cap layer was deposited on the epi-MgO by the DC reactive sputtering technology. Finally, a home-made metal organic chemical vapor deposition system was used to deposit YBCO films on the LMO/epi-MgO/IBAD-MgO/SDP-Y_2_O_3_ buffer layer. The detailed YBCO deposition process can be seen elsewhere [1,9].

The biaxial texture was characterized in situ using high-energy electron diffraction (RHEED) equipment. The picture exhibited regular diffraction spots array with biaxial textured MgO films. RHEED involves emitting a beam of high-energy electrons (5~100 keV) from an electron gun, which is incident on the sample surface at a small grazing angle (1~5°) to generate an electron diffraction beam. The crystal structure was then displayed by collecting the signal on a fluorescent screen. The RHEED came in at a shallow incident angle which was perpendicular to the plane of the ion gun. The diffracted pattern was incident on a phosphorescent screen and the resulting image was then collected by a charge coupled device (CCD) camera, which was interfaced to a computer. The electron beam evaporator was placed off-center to allow the ion gun full angular range. The substrate was rotatable azimuthally about the substrate normal. 

X-ray diffraction (XRD) (Bede D1) was used to probe the orientation and microstructure of the MgO films. We used symmetric scan geometry for the majority of the XRD measurements. The term symmetric scan kept the orientation of the scattering vector (q) fixed relative to the sample. In more practical terms, the inclined angle *θ* and 2*θ* were locked and changed in a ratio of 1:2, respectively. The magnitude of q varied as the angles changed. For the *θ*–2*θ* scan geometry, the change in angle corresponded to a change in the lattice spacing and both the orientation and phase of a sample can be observed in the collected scan. The detailed description of the *θ*–2*θ* scan can be seen elsewhere [41]. 

It is also necessary to utilize *ω*-scan and *Φ*-scan to obtain the texturing information, such as out-of-plane texture and in-plane texture. ω-scan is mainly used to determine the degree of crystallinity ordering in the films. During the ω-scan, the receiver was fixed at the 2*θ* position of the desired crystal face of the film, such as the (002) diffraction peak position of MgO. Subsequently, the sample stage rotated around an angle as the central angle, testing the angle range. Through computer fitting of the ω-scan curve, the half-maximum width (FWHM) of the out-of-plane texture was obtained. The lower the FWHM value, the more ordered the crystal grains of this orientation were arranged. 

*Φ*-scan was mainly used to determine the degree of ordering of epitaxial thin films in the a-b plane. Before conducting *Φ*-scan, the material’s strongest relative peak intensity surface (002) was selected from the PDF card, assuming that the angle between this surface and the sample surface was φ. During testing, the sample rotated to an angle *Φ*, and then the sample stage and receiver were fixed at the *θ* and 2*θ* positions of the desired crystal face. The sample rotated around its normal direction, and through fitting of the *Φ*-scan curve, the half-maximum width of the in-plane texture was obtained. For example, a (220) phi scan of a single crystal of MgO rotated about the (200) surface normal would have four peaks from 0° to 360° of phi angle. This scan gave the distribution of orientations of crystallites relative to one another in-plane. The FWHM value can be used as a measure and its value was used to characterize the goodness of the in-plane texture. Contributions to the width of a high-quality single crystal were the result of the instrument broadening as these widths should be very near 0°. Finally, the surface morphologies of the epi-MgO films were analyzed using the SPM 300HV scanning probe platform from Seiko and the INSPECT F50 SEM platform.

## 3. Results

The growth temperature plays a crucial role in the epitaxial growth of epi-MgO films, influencing both the structure and surface morphology of the films. Figure 1 displays the XRD *θ*–2*θ* diffraction patterns of MgO films, which were deposited at different temperatures with a film thickness of 250 nm, a deposition rate of 1.2 nm/s, and oxygen pressure of 10^−2^ Pa. The intensity of the MgO (002) diffraction peak was weak at the deposition temperature of 150 °C, indicating that this temperature failed to provide sufficient migration energy for the Epi-MgO films to grow along the *c*-axis orientation. The MgO migration energy increased with the deposition temperature increasing, resulting in a gradual increase in the peak intensity of the (002) orientation of the film. However, the maximum temperature that the heater could reach was 500 °C in the electron beam evaporation system. Therefore, the research was stopped at 500 °C.

Figure 2 illustrates the variations of the out-of-plane and the in-plane FWHM values of the epi-MgO films as a function of the deposition temperature. As shown in the figure, the deposition temperature influenced the biaxial texture of the films significantly. The out-of-plane and in-plane FWHM values of the films are 4.6° and 8° at a deposition temperature of 150 °C, respectively.

As the deposition temperature gradually increased, both the out-of-plane and in-plane textures were gradually optimized. When the deposition temperature reached 500°C, the biaxial texture of the epi-MgO films was Δω = 2.2° and ΔΦ = 4.8°, respectively. It has been proposed that the grain size of MgO increases with the deposition temperature [42]. The large self-epitaxial MgO grains can release the stress in the MgO films between the interface of IBAD-MgO and epi-MgO [43]. Therefore, the epi-MgO films can obtain better biaxial texture at the higher deposition temperature. The biaxial texture can be comparable to those from other researchers [38,39,40].

In order to investigate the effect of oxygen pressures on the MgO films, the epi-MgO films were prepared under different oxygen pressures while keeping other parameters constant with a deposition thickness of 150 nm, the growth temperature of 300 °C, and deposition rate of 1 nm/s. Figure 3 shows the in-plane and out-of-plane FWHM results of the MgO films prepared under different oxygen pressures. We found out that the oxygen pressure had little effect on the biaxial texture of MgO. 

The variation in the root mean square roughness (RMS) of the film surface with respect to the oxygen flow rate is illustrated in Figure 4. The RMS of the film surface gradually decreased from 9 nm to 4.5 nm as the oxygen flow increased from 0 to 3.5 × 10^−2^ Pa, indicating that as the oxygen flow rate increased, the oxygen defects in the film decreased, and the surface morphology was improved.

The deposition rate of the film also had a significant impact on the quality of the Epi-MgO films. By adjusting the electron evaporation’s beam current parameters, the film’s deposition rate was changed while keeping the film thickness, deposition temperature, and oxygen flow rate constant. The effect of the deposition rate on the film’s biaxial texture is shown in Figure 5.

The deposition rate had little effect on the biaxial texture. Subsequently, the surface morphology of these samples was characterized using SEM, and the results are shown in Figure 6. The film surface was smooth and dense with the deposition rates of 0.3 nm/s and 1 nm/s. When the deposition rate was increased to 3 nm/s, some particle agglomerations appeared on the surface. It has been shown that large particles on the film surface have an adverse effect on the superconducting properties of the subsequent YBCO film [44]. Therefore, the film’s deposition rate should not exceed 3 nm/s.

Thickness is one of the key growth parameters that affect the grain size, crystallinity, and surface morphology of MgO. Figure 7 shows the XRD *θ*–2*θ* scan results of the Epi-MgO films with a thickness range of 54–720 nm. The film thickness was controlled by the carrier frequency, while other deposition parameters were maintained at a deposition temperature of 450 °C, a deposition rate of 1.2 nm/s, and an oxygen flow rate of 1.6 × 10^−2^ Pa. It should be pointed out that IBAD MgO films are extremely thin (10 nm). The researchers in LANL found out that epi-MgO improved the in-plane texture significantly from IBAD-MgO [45]. They attributed the improvement in these films to the healing of the ion induced damage caused in the IBAD-MgO films. Therefore, the increase in the film thickness led to an enhancement in the crystallinity of MgO, resulting in an increase in the (002) diffraction peak intensity.

The variation in the FWHM values with different thicknesses in the MgO films was analyzed by the *ω*-scan and *φ*-scan, as presented in Figure 8, which shows that the in-plane and out-of-plane textures of the film gradually improved with the increase in the film thickness. In addition, the RMS roughness of the epi-MgO films varied with the thickness of the film, as shown in Figure 9, where the RMS roughness of the film surface increased with the increase in MgO thickness. These results indicate that the thicker MgO films result in an improvement of the biaxial texture but the deterioration of the surface morphology.

Figure 10a,b illustrate the RHEED diffraction patterns of the 10 nm-thick IBAD-MgO and 50 nm-thick epi-MgO, respectively. It can be observed that the diffraction spots of IBAD-MgO are larger than those of Epi-MgO, suggesting that the epi-MgO films improve the biaxial texture of IBAD-MgO significantly.

Since the lattice mismatch between MgO and the YBCO is relatively high (~8.6%), it is necessary to deposit a template layer between MgO and YBCO. The lattice mismatch between LMO and YBCO is low (~0.8%), which reduces the influence of buffer layer on YBCO growth during the epitaxy process. Subsequently, the YBCO functional layer was deposited on the LMO layer. Figure 11 shows the XRD *θ*–2*θ* scan of the YBCO films deposited on the LMO/epi-MgO/IBAD-MgO/Y_2_O_3_ buffer layers using metal-organic vapor deposition. The diffraction pattern shows that YBCO was growing along the pure c-axis and the (001) oriented peak intensity was strong. Nevertheless, the Y_2_O_3_ films exhibited a weak (004) peak, indicating that the Y_2_O_3_ films crystallized in the buffer layers during the YBCO process. It can be seen that the Y_2_O_3_ films were completely amorphous within the MgO films in Figure 7 because no Y_2_O_3_ diffraction peaks can be found in the *θ–2θ* scans. It should be mentioned that the deposition temperature of YBCO reached 700 °C, which can be seen in our previous research. The crystallization temperature of Y_2_O_3_ was 500 °C. Therefore, the Y_2_O_3_ films were re-crystallized during the YBCO films.

We used the four-probe method to test the superconductivity of the YBCO films. Figure 12 shows the critical current of the YBCO films prepared on the LMO/epi-MgO/IBAD-MgO buffer layer. The width of the metal template was 10 mm. However, this work could not directly measure the *Ic* value of the 10 mm-wide YBCO tape because of the limitation of current carrying capacity from our current source. The silver array was deposited on the YBCO tapes as a conducting electrode. The distance between the adjacent silver electrodes was 0.3 mm. The critical current value could be tested by mounting the test probe onto the adjacent silver electrode. Assuming the YBCO films were uniformly distributed over the 10 mm-wide tapes, the performance of the superconducting strip can divide the *Ic* of the 10 mm-wide YBCO tape by the *Ic* of the 0.3 mm-wide YBCO tape. The thickness of the YBCO was 1µm and the *Ic* value was tested under the condition of self-field. It can be seen that the critical current along the silver electrode is 9.27 A/0.3 mm in Figure 12. It can be calculated that the total critical current is 301 A/cm. After further calculation, the critical current density was 6 MA/cm^2^ (77 K, 500 nm, self-field). The above research shows that high-quality YBCO thin films can be epitaxially grown on the LMO/epi-MgO/IBAD-MgO template.

## 4. Conclusions

This paper investigates the influence of fabrication parameters, such as deposition temperature, film thickness, deposition rate, and oxygen partial pressure, on the structure, texture, and surface morphology of epi-MgO films during the preparation process using IBAD-MgO films as substrates. It was found that the crystalline quality of the epi-MgO films was improved when the deposition temperature was increased from 150 °C to 500 °C. The out-of-plane FWHM reduced from 4.2° to 2.2°, and the in-plane FWHM decreased from 8° to 4.8°. When the film thickness ranged from 54 nm to 720 nm, the RMS value rapidly rose from 1.6 nm to 25 nm and the in-plane texture increased from 5.4° to 2.2°. The oxygen pressure and deposition rate have a minimal impact on the biaxial texture but significantly affect its surface morphology. The critical current density of the YBCO films deposited on the LMO/epi-MgO/IBAD-MgO template was 6 MA/cm^2^ (77 K, 500 nm, self-field), indicating that this research provides a high-quality substrate for the YBCO layer.

## Figures and Tables

**Figure 1 micromachines-14-01914-f001:**
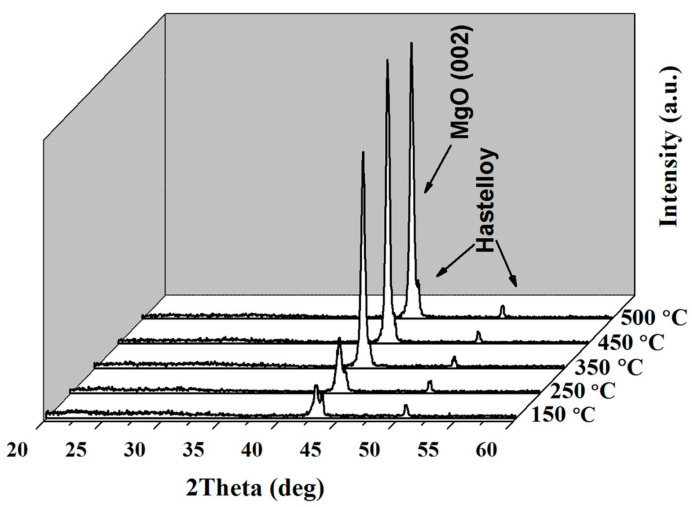
XRD *θ*–2*θ* diffraction results of epi-MgO deposited at different temperatures.

**Figure 2 micromachines-14-01914-f002:**
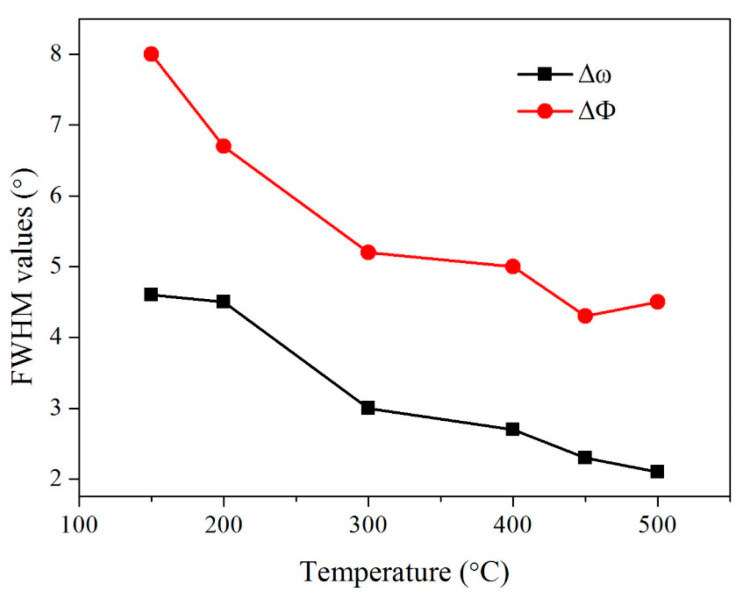
The FHWM values of Epi-MgO prepared at different deposition temperatures.

**Figure 3 micromachines-14-01914-f003:**
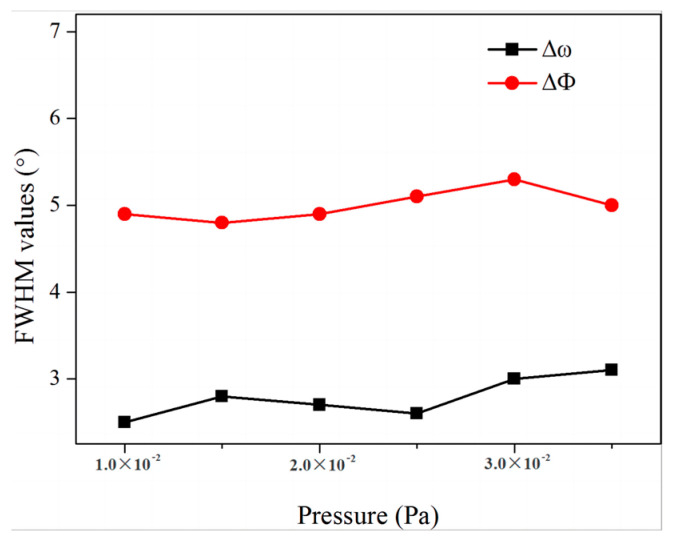
The FWHM values of in MgO films with different oxygen flux.

**Figure 4 micromachines-14-01914-f004:**
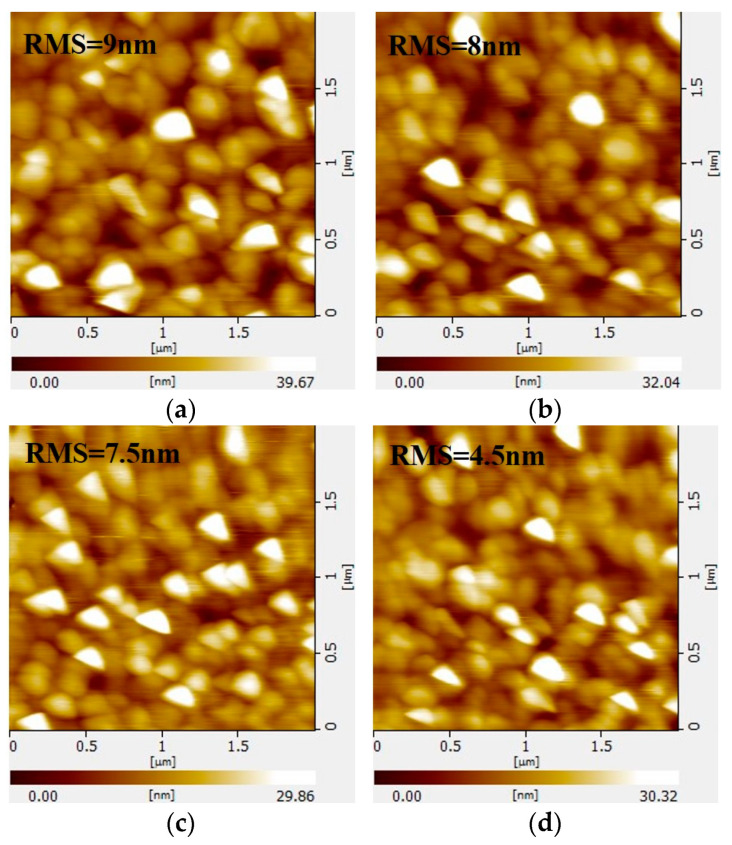
The RMS of MgO films with different oxygen flux: (**a**) 0; (**b**) 10^−2^ Pa; (**c**) 2.5 × 10^−2^ Pa; (**d**) 3.5 × 10^−2^ Pa.

**Figure 5 micromachines-14-01914-f005:**
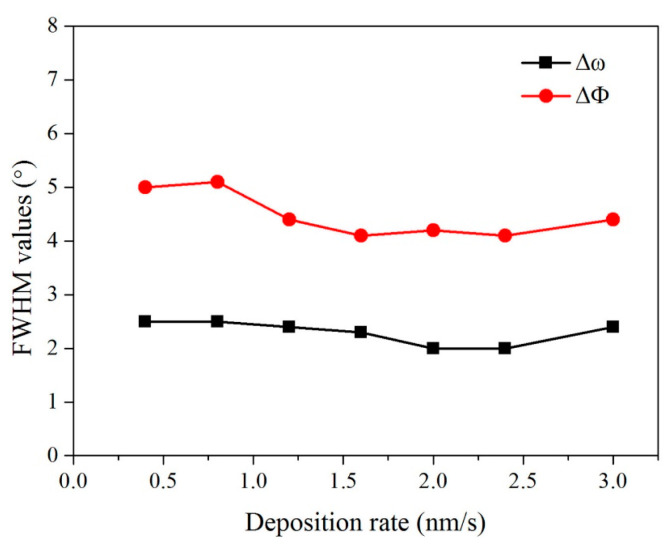
The texture changes in epi-MgO films with different deposition rates.

**Figure 6 micromachines-14-01914-f006:**

SEM images of Epi-MgO films with different deposition rates. (**a**) 0.3 nm/s; (**b**) 1 nm/s; (**c**) 3 nm/s.

**Figure 7 micromachines-14-01914-f007:**
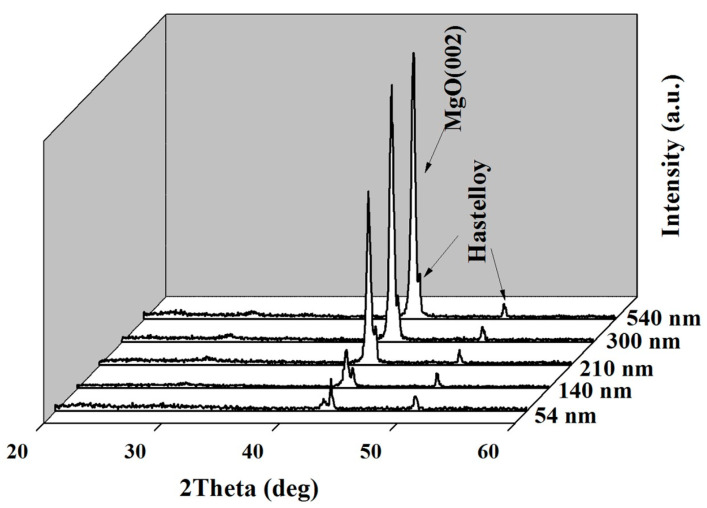
The XRD *θ*–2*θ* scans of the epi-MgO films with different thicknesses.

**Figure 8 micromachines-14-01914-f008:**
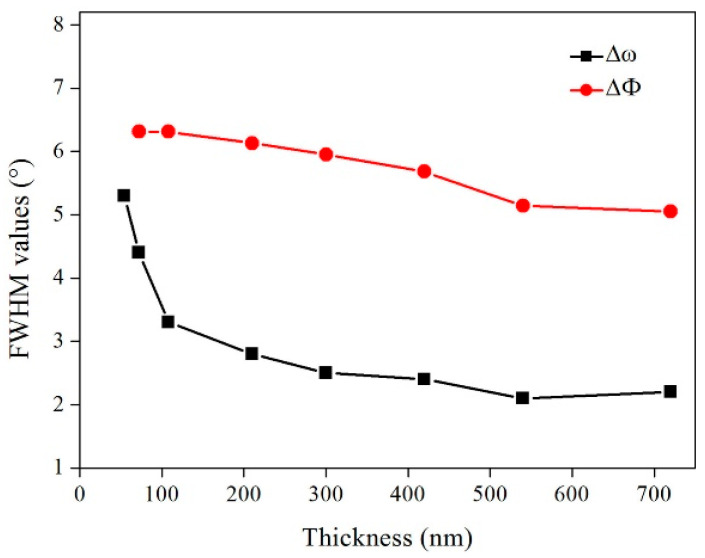
The in-plane out-plane texture of epi-MgO films with different deposition thicknesses.

**Figure 9 micromachines-14-01914-f009:**
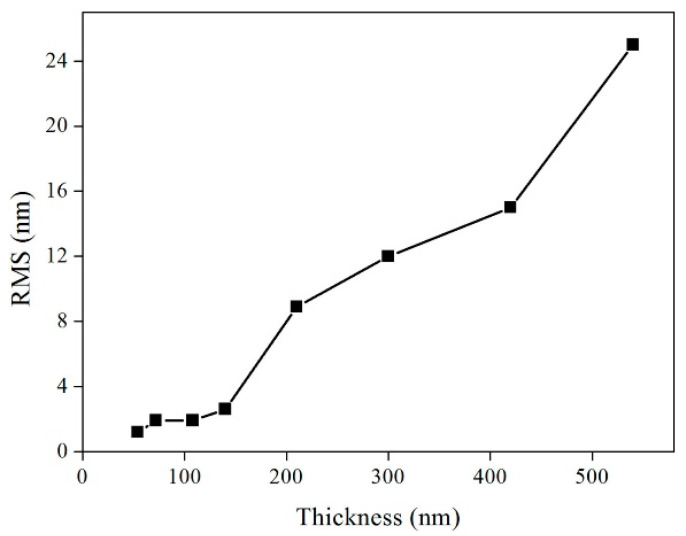
The surface root-mean-square roughness of epi-MgO films varies with different deposition thicknesses.

**Figure 10 micromachines-14-01914-f010:**
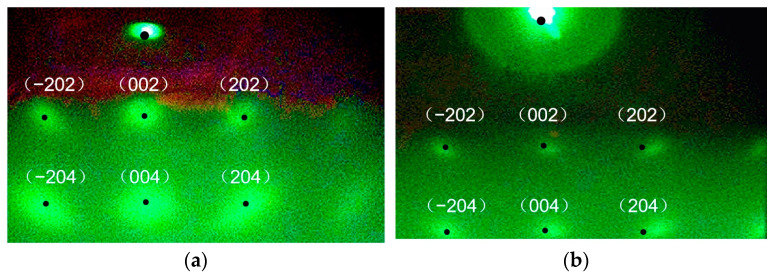
RHEED diffraction patterns of the films: (**a**) IBAD-MgO; (**b**) Epi-MgO.

**Figure 11 micromachines-14-01914-f011:**
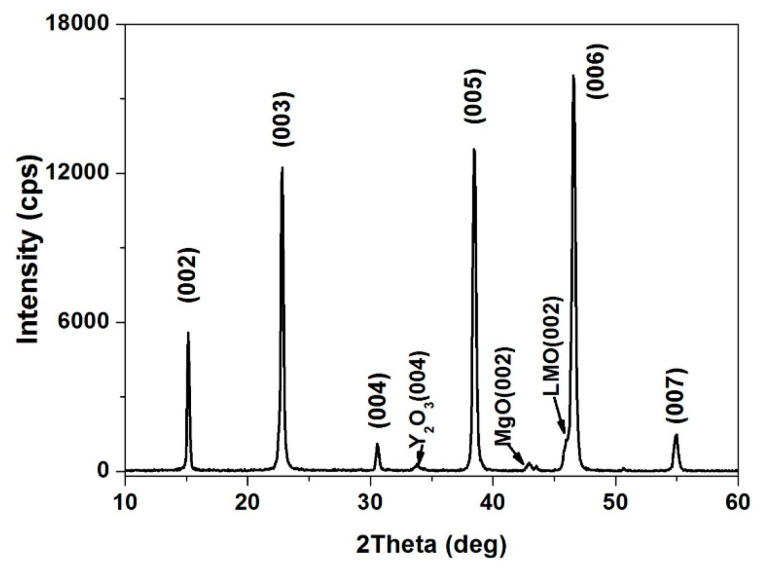
The XRD *θ*–2*θ* scan of YBCO layer.

**Figure 12 micromachines-14-01914-f012:**
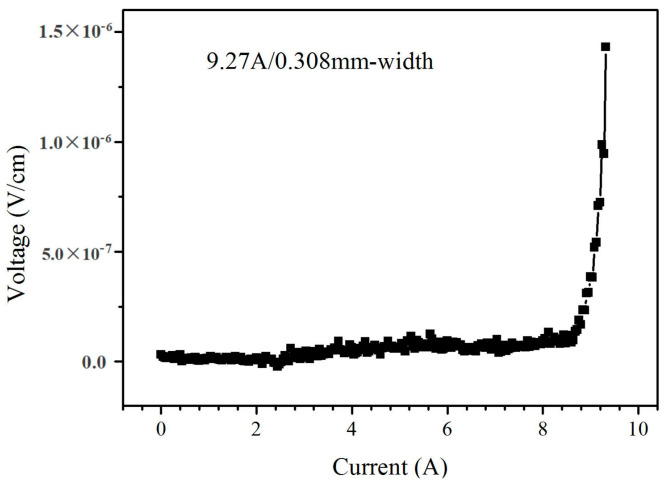
The I-V characteristic curve of YBCO film under 77 K and self-field conditions.

## Data Availability

All data generated or analyzed during this study are included in this article.

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
