# Peer review of "Preparation of MgO Self-Epitaxial Films for YBCO High-Temperature Coated Conductors"

_micromachines, 2023, doi:10.3390/mi14101914_

Round 1

Reviewer 1 Report

The author deposited homogeneous epitaxy MgO layer on IBAD-MgO to improve the biaxial texture for YBCO coated conductors. After the process research, the best full width at half maximum (FWHM) values are 2.2° for the out-of-plane texture and 4.8° for the in-plane texture, respectively. The biaxial textures are quite well for YBCO. It is original and innovative in the field of YBCO. The paper should be published in micromachines. But before the publication, there has some deficiency need to be revised in new version.

1. The best texture is 2.4° (out-of-plane) and 4.8° (in-plane) for MgO films. How about of the research status from other researchers. I suggest the author to cite more references

2. Please explain why the Y2O3 (004) peak appeared in figure 11

 Minor editing of English language required

Author Response

Revisions in response to the Editors’ comments:

  1. Reviewer1 wrote:

The author deposited homogeneous epitaxy MgO layer on IBAD-MgO to improve the biaxial texture for YBCO coated conductors. After the process research, the best full width at half maximum (FWHM) values are 2.2° for the out-of-plane texture and 4.8° for the in-plane texture, respectively. The biaxial textures are quite well for YBCO. It is original and innovative in the field of YBCO. The paper should be published in micromachines. But before the publication, there has some deficiency need to be revised in new version.

  1. The best texture is 2.4° (out-of-plane) and 4.8° (in-plane) for MgO films. How about of the research status from other researchers. I suggest the author to cite more references

Our Response:

We greatly appreciate this question, which help us further improve paper quality. We totally agree with the suggestion and increase the recent advances in IBAD-MgO as follow:

Therefore, the epi-MgO films can obtain better biaxial texture at the higher deposition temperature. The biaxial texture is much better than those from other researchers [38-42].

Location of Change: Page 5    line 178-179

  1. Please explain why the Y2O3 (004) peak appeared in figure 11

Our Response:

We greatly appreciate this question, which help us further improve paper quality. We totally agree with the suggestion and increase the recent advances in IBAD-MgO as follow:

Nevertheless, the Y2O3 films exhibit a weak (004) peak, indicating that Y2O3 films have been crystallized in the buffer layers during the YBCO process. It can be seen that the Y2O3 films are completely amorphous during the MgO films in Figure 7 because no Y2O3 diffraction peaks can be found in the q-2q scans. It should be mentioned that the deposition temperature of YBCO reach 700°C, which can be seen in our previous research. The crystallization temperature of Y2O3 is 500°C. Therefore, the Y2O3 films are re-crystallized during the YBCO films.

Location of Change: Page 8    line 251-257

Reviewer 2 Report

I have carefully read the paper and I must say the paper quality is very poor, with many critical aspects that would require a deep amendment.

 But what is more important, this work lacks of any originality.

 It is entirely focused on the validation of IBAD MgO-Epi MgO as template for the growth of superconducting REBCO layer.

This is something perfectly known to anyone working on REBCO tapes and coated-conductor technology.

 The first use of IBAD MgO and the possibility to have textured film on an amorphous template by the assistance of an ion beam was reported by Bob Hammond from Stanford University in 1982!

And the first application in REBCO tapes dates back in 1995 by Hammond and K.B. Do.

Since then, many studies have been performed and published on the IBAD MgO – Epi MgO structure, in particular when grown on Y2O3 amorphous template, and having LMO on top as a good template for REBCO films.

This is exactly what this paper is on, and authors also state they have proposed this architecture. They didn’t propose that, this is the “standard” for the coated-conductor community.

As a confirmation, most of the companies currently commercializing REBCO tapes use this kind of architecture.

 Therefore, it is my opinion that this paper does not deserve to be published.

English can be improved, even if the general sense of the paper and of the sentences can be easily understood. 

Author Response

Revisions in response to the Editors’ comments:

  1. Reviewer2 wrote:

I have carefully read the paper and I must say the paper quality is very poor, with many critical aspects that would require a deep amendment. But what is more important, this work lacks of any originality. It is entirely focused on the validation of IBAD MgO-Epi MgO as template for the growth of superconducting REBCO layer. This is something perfectly known to anyone working on REBCO tapes and coated-conductor technology. The first use of IBAD MgO and the possibility to have textured film on an amorphous template by the assistance of an ion beam was reported by Bob Hammond from Stanford University in 1982! And the first application in REBCO tapes dates back in 1995 by Hammond and K.B. Do. Since then, many studies have been performed and published on the IBAD MgO – Epi MgO structure, in particular when grown on Y2O3 amorphous template, and having LMO on top as a good template for REBCO films. This is exactly what this paper is on, and authors also state they have proposed this architecture. They didn’t propose that, this is the “standard” for the coated-conductor community. As a confirmation, most of the companies currently commercializing REBCO tapes use this kind of architecture.

Our Response:

We greatly appreciate this question, which help us further improve paper quality. We totally agree with the suggestion and increase the recent advances in IBAD-MgO as follow:

Yttria Stabilized Zirconia (YSZ) (ZrO2–Y2O3 stabilized) is one of the earliest materials to be used as biaxially textured template for YBCO. YSZ is cubic and has a lattice constant of 5.14 Aº. In 1992, Iijima et al. reported that a sharp biaxially aligned structure of YSZ films grown on alloy as a substrate could be developed by IBAD technology [32]. The YSZ films can be successfully used for YBCO films and resulted in demonstration of high Jc values [33]. The in-plane FWHM of the XRD in-plane scan range from 7-15° [34-36]. The most significant disadvantage in using YSZ as a template material is that the texture evolution occurs rather slowly, requiring about 1 µm film thickness to achieve the optimum in-plane texture [35]. This requires a deposition time of several hours to achieve the desired thickness of the template material. This has limited YSZ as a texturing material in many applications.

The biaxial texture formation of Magnesium Oxide (MgO) grown by IBAD was first reported by the researchers from Stanford University [37]. In contrast to IBAD-YSZ, IBAD grown MgO has been observed to have optimum biaxial texture at about 10 nm of film thickness, while IBAD-YSZ requires typically about 700-1000nm thickness of film. From a practical point of view, IBAD-MgO growth is about 100 times faster than that for IBAD-YSZ, which translates to significantly lower costs. Current work has been focused on production of long lengths of IBAD-MgO. The textured MgO template has been adopted world-wide as the technology of choice from the USA to Japan to Korea [11,15,32,36]. However, improvements in the quality of IBAD-MgO have seemingly stalled. The standard IBAD-MgO template is composed of LaMnO3 (LMO)/epitaxial-MgO/IBAD-MgO/ amorphous-Y2O3/amorphous-Al2O3 five-layer stack. In our previous study, the solution deposition planarization (SDP) method, which reduced the preparation cost and process complexity of YBCO superconducting strip, was used to deposit oxide amorphous films on the surface of Hastelloy C276 tapes [38]. The SDP-Y2O3 layer can replace the three processes, including electrochemical polishing, sputtering-Al2O3 and sputtering-Y2O3 in the commercialized buffer layer.

In our previous research, LaMnO3 (LMO)/IBAD-MgO/SDP-Y2O3 has been used to server the buffer template for YBCO films [38]. However, the buffer layer shows poor biaxial texture with the out-of-plane the full width at half maximum (FWHM) value of 3.5° and in-plane FWHM value of 7.2° in the buffer layer, respectively. It should be noted that sufficient critical currents can hardly be obtained with the in-plane texture over 5° [39]. Therefore, it’s necessary to reduce the biaxial texture. Y. Yamada et al. have deposited CeO2 on IBAD-MgO to improve the biaxial texture. However, the best in-plane FWHM value is 6° in their research [40].

Location of Change: Page 2    line 56-90

Reviewer 3 Report

The main note about the article is a very brief description of the methods. In addition to SPM and SEM, information is needed on XRD, electron beam evaporation and measurement of the thickness and growth rate. It is required to give details of thickness measurement using RHEED and about control of the growth rate (see also line 162, frequency of what).

In order for the article to be understandable not only to narrow specialists, detailed information is necessary about the substrate.

There are a number of inaccuracies (typos)
line 101 MgO atoms (?)
line 106 half height and width (?)
lines 119-123 this paragraph should be moved to section 2
lines 164-165 "increase in film thickness leads to an enhancement of the crystallinity of MgO" - this statement requires clarification

In conclusion, the reviewer notes that he is not an expert in diffraction methods for analyzing the texture of thin films. Therefore, the article should be reviewed by an appropriate specialist after the authors give details of the X-ray methods used.

Author Response

Revisions in response to the Editors’ comments:

  1. Reviewer3 wrote:

The main note about the article is a very brief description of the methods. In addition to SPM and SEM, information is needed on XRD, electron beam evaporation and measurement of the thickness and growth rate. It is required to give details of thickness measurement using RHEED and about control of the growth rate (see also line 162, frequency of what).

In order for the article to be understandable not only to narrow specialists, detailed information is necessary about the substrate.
There are a number of inaccuracies (typos)
line 101 MgO atoms (?)
line 106 half height and width (?)
lines 119-123 this paragraph should be moved to section 2
lines 164-165 "increase in film thickness leads to an enhancement of the crystallinity of MgO" - this statement requires clarification

Our Response:

We greatly appreciate this question, which help us further improve paper quality. We totally agree with the suggestion and increase the recent advances in IBAD-MgO as follow:

The MgO migration energy increases with the deposition temperature increasing,

Location of Change: Page 4    line 160-161

Figure 2. The FHWM values of Epi-MgO prepared at different deposition temperatures

Location of Change: Page 4    line 166

It is also necessary to utilize ω-scan and Φ-scan to obtain the texturing information, such as out-of-plane texture and in-plane texture. ω-scan is mainly used to determine the degree of crystallinity ordering in the films. During the ω-scan, the receiver is fixed at the position of the desired crystal face of the film, such as the (002) diffraction peak position of MgO. Subsequently, the sample stage rotates around an angle as the central angle, testing the angle range. Through computer fitting of the ω-scan curve, the half-maximum width (FWHM) of the out-of-plane texture can be obtained. The lower the FWHM value, the more ordered the crystal grains of this orientation are arranged.

Location of Change: Page 3    line 130-137

It should be point out that IBAD MgO films are extremely thin films. The researchers in LANL found out that the epi-MgO improves the in-plane texture significantly from IBAD-MgO [44]. They attributed the improvement in these films to the healing of the ion induced damage caused in the IBAD-MgO films. Therefore, the increase in film thickness leads to an enhancement of the crystallinity of MgO, resulting in an increase in the (002) diffraction peak intensity.

Location of Change: Page 6,7    line 219-225

In conclusion, the reviewer notes that he is not an expert in diffraction methods for analyzing the texture of thin films. Therefore, the article should be reviewed by an appropriate specialist after the authors give details of the X-ray methods used. English can be improved, even if the general sense of the paper and of the sentences can be easily understood. 

Our Response:

We greatly appreciate this question, which help us further improve paper quality. We believe the English has been improved. We totally agree with the suggestion and increase the recent advances in IBAD-MgO as follow:

The crystal structure is then displayed by collecting the signal on a fluorescent screen. The RHEED comes in at a shallow incident angle which is perpendicular to the plane of the ion gun. The diffracted pattern is incident on a phosphorescent screen and the resulting image is then collected by a CCD camera, which is interfaced to a computer. The electron beam evaporator is placed off-center to allow the ion gun full angular range. The substrate is rotatable azimuthally about the substrate normal. Not shown is the QCM rate monitor and ion probe used to measure the electron beam evaporation rate and the ion beam current density, respectively.

When the metal oxide films are prepared using electron beam evaporation, the ionic bonds of the materials can be easily disrupted by the electron beam, potentially causing the composition of the film material to deviate and affecting the microstructure of the film. Therefore, it is necessary to introduce oxygen to maintain the proper proportion of MgO film composition during the epi-MgO deposition process.

X-ray diffraction (XRD) was used to probe the orientation and microstructure of MgO films. We used symmetric scan geometry for the majority of the XRD measurements. The term symmetric scan keeps the orientation of the scattering vector (q) fixed relative to the sample. In more practical terms, the inclined angle θ and are locked and changed in a ratio of 1:2, respectively. The magnitude of q varies as the angles are changed. For the θ-2θ scan geometry, the change in angle corresponds to a change in the lattice spacing and both the orientation and phase of a sample can be observed in the collected scan.

It is also necessary to utilize ω-scan and Φ-scan to obtain the texturing information, such as out-of-plane texture and in-plane texture. ω-scan is mainly used to determine the degree of crystallinity ordering in the films. During the ω-scan, the receiver is fixed at the position of the desired crystal face of the film, such as the (002) diffraction peak position of MgO. Subsequently, the sample stage rotates around an angle as the central angle, testing the angle range. Through computer fitting of the ω-scan curve, the half-maximum width (FWHM) of the out-of-plane texture can be obtained. The lower the FWHM value, the more ordered the crystal grains of this orientation are arranged.

Φ-scan is mainly used to determine the degree of ordering of epitaxial thin films in the a-b plane. Before conducting Φ-scan, the material's strongest relative peak intensity surface is selected from the PDF card, assuming that the angle between this surface and the sample surface is φ. During testing, the sample rotates to an angle Φ, and then the sample stage and receiver are fixed at the θ and positions of the desired crystal face. The sample rotates around its normal direction, and through fitting of the Φ-scan curve, the half-maximum width of the in-plane texture can be obtained. For example, a (220) phi scan of a single crystal of MgO rotated about the (200) surface normal would have four peaks from 0° to 360° of phi angle. This scan gives the distribution of orientations of crystallites relative to one another in-plane. The FWHM value can be used as a measure and its value is used to characterize the goodness of the in-plane texture. Contributions to the width of a high-quality single crystal are the result of instrument broadening as these widths should be very near 0°.

Location of Change: Page 3    line 110-150

Round 2

Reviewer 2 Report

General remarks

The study is well done and presented. Authors have investigated the influence of almost all the growth process parameters on the epi-MgO layer quality.

I’d like the authors to clearly state that these results are for an MgO bilayer (IBAD and epi-) grown on SDP yttria layer. This can have a strong influence on the obtained and presented results. Repeating the same study on the standard amorphous Al2O3/Y2O3 structure could give different results. And, in my opinion, this is also one of the strongest point of this manuscript.

For example, it is never mentioned in the abstract. It definitely should be.

Abstract

The biaxial texture can be optimized by the deposition of additional epi-MgO layer.

This sentence is kind of isolated from the remaining text.

I suggest to join this sentence and the previous one in a single one.

After the research of epi-MgO process…

Do the authors mean, the best results?

Introduction

In order to realize the application of high-temperature superconducting materials in the field of strong electricity, the critical current (Ic) must be as high as 200A/cm [7]

I can’t understand what “strong electricity” means. Moreover, the Ic threshold for applications depends on the applications and, first of all, from the operative conditions. 200 A/cm at 4.2 K self field, for example, is way lower than what required.

Moreover, the YBCO conductors must be epitaxial growth on cheap metal substrates by the thin film deposition method to increase the mechanical properties [8]

Replace with “Epitaxial grown

Epitaxial growth is required to avoid grain boundaries misalignment and metallic templates are more for intended for giving flexibility. Mechanical properties are too generic.

The functions of the buffer layers can be described as follows: Firstly, block the mutual diffusion of elements between the metal substrate and the YBCO layer. The flat and dense oxide buffer films can prevent direct contact between the superconducting layer and the YBCO layer, effectively blocking the mutual diffusion of elements [15-17]

Remove capital letter from “First”.

between the superconducting layer and the YBCO layer” YBCO is the superconducting layer. I would use REBCO rather than YBCO.

Finally, the biaxial texture required for YBCO growth can be introduced by the film epitaxy method in the oxide buffer layers [21-23]

This sentence is not clear. The biaxial texture is “induced” in the YBCO layer by epitaxial growth on a textured “oxide buffer layers.

“At present, there are three well explored technologies for obtaining biaxial texture, i.e., rolling-assisted biaxial texture, inclined substrate deposition and ion beam-assisted deposition (IBAD) [24].”

Please use the acronyms also for the other techniques (RABiTS and ISD)

The standard IBAD-MgO template is composed of LaMnO3 (LMO)/epitaxial-MgO/IBAD-MgO/ amorphous-Y2O3/amorphous-Al2O3 five-layer stack.

I would not call that architecture “The standard IBAD-MgO template” but rather “the standard coated-conductor buffer structure using IBAD-MgO”.

However, the buffer layer shows poor biaxial texture with the out-of-plane the full width at half maximum (FWHM) value of 3.5° and in-plane FWHM value of 7.2° in the buffer layer, respectively”

Authors must define better what they are talking about. FWHM regards the XRD rocking curve of some reflection, and the in-plane is about the f-scan.

Experiment

The MgO homoepitaxial layer was deposited on the substrate using electron beam evaporation from  a crucible

This has been already told the sentence before.

Please define “QCM

the material's strongest relative peak intensity surface is selected from the PDF card

What “surface” is for in this sentence?

Results

Can the author find a maximum temperature after which the intensity of the MgO (002) peak and/or the biaxial texture start to degrade? If not, why did they stop the study at 500 °C?

In Figure 1, please put the temperatures text on the right of the spectra to make them more readable.

The biaxial texture is much better than those from other researchers [38-42]”. Please rephrase this sentence in a more polite way.

In order to investigate the effect of oxygen flow rate on the MgO films, the epi-MgO films were prepared under different oxygen pressures

Flow and pressure are two distinct parameters and may have different influence. Authors are studying here the influence of pressure, which can be dynamically kept with different gas flows, depending on the gas outlet aperture. Probably, in their case the aperture is always the same, so on changing the flow also the pressure changes. But these are two distinct physical quantities.

The RMS of the film surface gradually decreases from 9nm to 4.5nm as the oxygen

Please insert a blank space between numbers and units in all the manuscript.

The oxygen flow rate has little effect on the biaxial texture. Subsequently, the surface morphology of these samples was characterized using SEM, and the results are shown in Figure 6

I presume here authors are talking about variations with “deposition rates” and not “oxygen flows”. I understand that the oxygen flow (better pressure, see my comment above) has an influence on the deposition rate. But which is the parameter governing the mechanism giving rise to different morphologies? The pressure or the growth rate? I guess the second one. Therefore, this sentence may result misleading.

Figure 7: as in figure 1, please put the thicknesses on the right of the spectra to make them more readable.

It should be point out that…

Please replace with “pointed out

We use the four-probe method to test the superconductivity of the YBCO films, Figure 12 shows the critical current test diagram of the YBCO film prepared on the LMO/epi-MgO/IBAD-MgO buffer layer. The width of the metal template is 10mm. However, this work can not directly measure the Ic value of the 10mm-wide YBCO tape because of the limitation of current carrying capacity from our current source. The silver electrode can be deposited on the YBCO surface, and the critical current value can be tested. The performance of the 1cm wide superconducting strip is then calculated

This sentence needs to be rephrased. For example, what is a “test diagram??”

What “the critical current of YBCO films along the silver electrode (0.308mm-width)” means?

I guess silver has been deposited to improve the electrical contacts in the current bias regions and as voltage taps to reduce the noise. But the I-V curve is not measured “along” the silver “electrodes

In general, these last two sections regarding Ic measurements must be amended.

Please provide the most important information first: film thickness, strip width (where the Ic was actually measured), Electric field criterion used to determine the Ic value, measured Ic value and evaluated Jc. Everything else is less important.

Conclusions

When the film thickness from 54 nm to 720nm, The RMS value rapidly rises from 1.6nm to 25nm"”

Please remove the capital “T” marked in red.

No relevant comments

Author Response

Dear Editor

Thank you so much for reviewing our work. We are delighted to be informed with a major review. The reviewers’ suggestions are very helpful to improve our manuscript. Here we submit our revised manuscript (the changes are highlighted in yellow) and point-to-point responses. We believe that we have improved the English writing.

Sincerely,

Yan Xue

Revisions in response to the Editors’ comments:

  1. Reviewer2 wrote:

The study is well done and presented. Authors have investigated the influence of almost all the growth process parameters on the epi-MgO layer quality.

I’d like the authors to clearly state that these results are for an MgO bilayer (IBAD and epi-) grown on SDP yttria layer. This can have a strong influence on the obtained and presented results. Repeating the same study on the standard amorphous Al2O3/Y2O3 structure could give different results. And, in my opinion, this is also one of the strongest point of this manuscript.

For example, it is never mentioned in the abstract. It definitely should be.

Abstract

 “The biaxial texture can be optimized by the deposition of additional epi-MgO layer.”

This sentence is kind of isolated from the remaining text.

I suggest to join this sentence and the previous one in a single one.

“After the research of epi-MgO process…”

Our Response:

We greatly appreciate this question, which help us further improve paper quality. We totally agree with the suggestion and increase the recent advances in epi-MgO as follow:

Ion beam-assisted deposition (IBAD) has been proposed as a promising texturing technology that uses film epitaxy method to obtain biaxial texture on a non-textured metal or compound substrate. Magnesium oxide (MgO) is the most well explored texturing material. In order to obtain the optimal biaxial texture, the actual thickness of the IBAD-MgO film must be controlled within 12nm. Due to the bombardment of ion beams, IBAD-MgO has large lattice deformation, poor texture and many defects in the films. In this work, the solution deposition planarization (SDP) method was used to deposit oxide amorphous Y2O3 films on the surface of Hastelloy C276 tapes instead of the electrochemical polishing, sputtering-Al2O3 and sputtering-Y2O3 in the commercialized buffer layer. An additional homogeneous epitaxy MgO (epi-MgO) layer, which was used to improve the biaxial texture in the IBAD-MgO layer, has been deposited on the IBAD-MgO layer by electron-beam evaporation. The effects of growth temperature, film thickness, deposition rate and oxygen pressure on the texture and morphology of the epi-MgO film were systematically studied. The best full width at half maximum (FWHM) values were 2.2° for the out-of-plane texture and 4.8° for the in-plane texture for epi-MgO films, respectively. Subsequently, the LaMnO3 cap layer and YBa2Cu3O7-x (YBCO) functional layer was deposited on the epi-MgO layer to test the quality of MgO layer. Finally, the critical current density of YBCO films is 6 MA/cm2 (77 K, 500 nm, self-field), indicating that this research provides a high-quality MgO substrate for the YBCO layer.

Location of Change: Page 1    line 11-27

  1. Reviewer2 wrote:

Introduction

In order to realize the application of high-temperature superconducting materials in the field of strong electricity, the critical current (Ic) must be as high as 200A/cm [7]

I can’t understand what “strong electricity” means. Moreover, the Ic threshold for applications depends on the applications and, first of all, from the operative conditions. 200 A/cm at 4.2 K self field, for example, is way lower than what required.

Moreover, the YBCO conductors must be epitaxial growth on cheap metal substrates by the thin film deposition method to increase the mechanical properties [8]

Replace with “Epitaxial grown

Epitaxial growth is required to avoid grain boundaries misalignment and metallic templates are more for intended for giving flexibility. Mechanical properties are too generic.

The functions of the buffer layers can be described as follows: Firstly, block the mutual diffusion of elements between the metal substrate and the YBCO layer. The flat and dense oxide buffer films can prevent direct contact between the superconducting layer and the YBCO layer, effectively blocking the mutual diffusion of elements [15-17]

Remove capital letter from “First”.

between the superconducting layer and the YBCO layer” YBCO is the superconducting layer. I would use REBCO rather than YBCO.

Finally, the biaxial texture required for YBCO growth can be introduced by the film epitaxy method in the oxide buffer layers [21-23]

This sentence is not clear. The biaxial texture is “induced” in the YBCO layer by epitaxial growth on a textured “oxide buffer layers.

“At present, there are three well explored technologies for obtaining biaxial texture, i.e., rolling-assisted biaxial texture, inclined substrate deposition and ion beam-assisted deposition (IBAD) [24].”

Please use the acronyms also for the other techniques (RABiTS and ISD)

The standard IBAD-MgO template is composed of LaMnO3 (LMO)/epitaxial-MgO/IBAD-MgO/ amorphous-Y2O3/amorphous-Al2O3 five-layer stack.

I would not call that architecture “The standard IBAD-MgO template” but rather “the standard coated-conductor buffer structure using IBAD-MgO”.

However, the buffer layer shows poor biaxial texture with the out-of-plane the full width at half maximum (FWHM) value of 3.5° and in-plane FWHM value of 7.2° in the buffer layer, respectively”

Authors must define better what they are talking about. FWHM regards the XRD rocking curve of some reflection, and the in-plane is about the f-scan.

Our Response:

We greatly appreciate this question, which help us further improve paper quality. We totally agree with the suggestion and increase the recent advances in epi-MgO as follow:

REBa2Cu3O7-x (REBCO) superconducting conductors exhibit excellent electrical properties and broad application prospects in the fields of transmission cables, strong magnets, motors, and current limiters [1-6]. In order to realize the application of high-temperature superconducting materials in the field of transmission cables, the critical current (Ic) must be as high as 200 A/cm at 77 K self-field [7]; Therefore, the REBCO conductors must be epitaxial grown on the metal substrates with biaxial texture by the thin film deposition method to avoid grain boundaries misalignment [8].

Location of Change: Page 1    line 30-40

The functions of the buffer layers can be described as follows: First, block the mutual diffusion of elements between the metal substrate and the REBCO layer. The flat and dense oxide buffer films can prevent direct contact between the superconducting layer and the REBCO layer, effectively blocking the mutual diffusion of elements [15-17]. Second, the adhesion of YBCO layer deposited on oxide buffer layers is much better than that on the metal substrates [18-20]. Finally, the biaxial texture is induced in the REBCO layer by the epitaxial grown on a textured oxide buffer layers [21-23].

Location of Change: Page 1~2    line 43-49

At present, there are three well explored technologies for obtaining biaxial texture, i.e., rolling-assisted biaxial texture (RABiTS), inclined substrate deposition (ISD) and ion beam-assisted deposition (IBAD) [24]

Location of Change: Page 2    line 50-52

The standard coated-conductor buffer structure using IBAD-MgO is composed of LaMnO3 (LMO)/epitaxial-MgO/IBAD-MgO/ amorphous-Y2O3/amorphous-Al2O3 five-layer stack.

Location of Change: Page 2    line 78-80

However, the buffer layer shows poor biaxial texture with the out-of-plane the full width at half maximum (FWHM) value of 3.5° (the rocking curve of LMO (002) peak) and in-plane FWHM value of 7.2° (the Φ-scan of LMO (222) peak) in the buffer layer, respectively.

Location of Change: Page 2    line 88-90

  1. Reviewer2 wrote:

Experiment

“The MgO homoepitaxial layer was deposited on the substrate using electron beam evaporation from  a crucible”

This has been already told the sentence before.

Please define “QCM”

“the material's strongest relative peak intensity surface is selected from the PDF card”

What “surface” is for in this sentence?

Our Response:

We greatly appreciate this question, which help us further improve paper quality. We totally agree with the suggestion and increase the recent advances in epi-MgO as follow:

The quartz crystal microbalance (QCM) rate monitor and ion probe are used to measure the electron beam evaporation rate and the ion beam current density, respectively.

Location of Change: Page 3    line 115-116

Before conducting Φ-scan, the material's strongest relative peak intensity surface (002) is selected from the PDF card, assuming that the angle between this surface and the sample surface is φ.

Location of Change: Page 3    line 151-152

  1. Reviewer2 wrote:

Results

Can the author find a maximum temperature after which the intensity of the MgO (002) peak and/or the biaxial texture start to degrade? If not, why did they stop the study at 500 °C?

In Figure 1, please put the temperatures text on the right of the spectra to make them more readable.

“The biaxial texture is much better than those from other researchers [38-42]”. Please rephrase this sentence in a more polite way.

“In order to investigate the effect of oxygen flow rate on the MgO films, the epi-MgO films were prepared under different oxygen pressures”

Flow and pressure are two distinct parameters and may have different influence. Authors are studying here the influence of pressure, which can be dynamically kept with different gas flows, depending on the gas outlet aperture. Probably, in their case the aperture is always the same, so on changing the flow also the pressure changes. But these are two distinct physical quantities.

“The RMS of the film surface gradually decreases from 9nm to 4.5nm as the oxygen”

Please insert a blank space between numbers and units in all the manuscript.

“The oxygen flow rate has little effect on the biaxial texture. Subsequently, the surface morphology of these samples was characterized using SEM, and the results are shown in Figure 6”

I presume here authors are talking about variations with “deposition rates” and not “oxygen flows”. I understand that the oxygen flow (better pressure, see my comment above) has an influence on the deposition rate. But which is the parameter governing the mechanism giving rise to different morphologies? The pressure or the growth rate? I guess the second one. Therefore, this sentence may result misleading.

Figure 7: as in figure 1, please put the thicknesses on the right of the spectra to make them more readable.

“It should be point out that…”

Please replace with “pointed out”

“We use the four-probe method to test the superconductivity of the YBCO films, Figure 12 shows the critical current test diagram of the YBCO film prepared on the LMO/epi-MgO/IBAD-MgO buffer layer. The width of the metal template is 10mm. However, this work can not directly measure the Ic value of the 10mm-wide YBCO tape because of the limitation of current carrying capacity from our current source. The silver electrode can be deposited on the YBCO surface, and the critical current value can be tested. The performance of the 1cm wide superconducting strip is then calculated”

This sentence needs to be rephrased. For example, what is a “test diagram??”

What “the critical current of YBCO films along the silver electrode (0.308mm-width)” means?

I guess silver has been deposited to improve the electrical contacts in the current bias regions and as voltage taps to reduce the noise. But the I-V curve is not measured “along” the silver “electrodes”

In general, these last two sections regarding Ic measurements must be amended.

Please provide the most important information first: film thickness, strip width (where the Ic was actually measured), Electric field criterion used to determine the Ic value, measured Ic value and evaluated Jc. Everything else is less important.

Our Response:

We greatly appreciate this question, which help us further improve paper quality. We totally agree with the suggestion and increase the recent advances in epi-MgO as follow:

However, the maximum temperature that the heater can reach is 500 °C in the electron beam evaporation system. Therefore, the research is stopped at 500 °C.

Location of Change: Page 4    line 173-175

Figure 1. XRD q-2q diffraction results of epi-MgO deposited at different temperatures

The biaxial texture can be comparable to those from other researchers [38-40].

Location of Change: Page 5    line 191-192

In order to investigate the effect of oxygen pressures on the MgO films

Location of Change: Page 5    line 193

The RMS of the film surface gradually decreases from 9 nm to 4.5 nm as the oxygen flow increases from 0 to 3.5x10-2 Pa, indicating that as the oxygen flow rate increases

Location of Change: Page 6    line 207-209

The deposition rate has little effect on the biaxial texture. Subsequently, the surface morphology of these samples was characterized using SEM, and the results are shown in Figure 6.

Location of Change: Page 7    line 221-223

It should be pointed out that IBAD MgO films are extremely thin (10 nm).

Location of Change: Page 7    line 232-233

Figure 7. The XRD q-2q scans of the epi-MgO films with different thicknesses

We use the four-probe method to test the superconductivity of the YBCO films. Figure 12 shows the critical current of the YBCO films prepared on the LMO/epi-MgO/IBAD-MgO buffer layer.

Location of Change: Page 9    line 271-273

The silver array has been deposited on the YBCO tapes as conducting electrode. The distance between the adjacent silver electrode is 0.3 mm. The critical current value can be tested by mounting the test probe onto the adjacent silver electrode. Assuming the YBCO films are uniformly distributed over the 10 mm-wide tapes, the performance of the superconducting strip can divide the Ic of the 10 mm-wide YBCO tape by the Ic of the 0.3 mm-wide YBCO tape. The thickness of YBCO is 1µm and the Ic value is tested under the condition of self-field. It can be seen that the critical current along the silver electrode is 9.27 A/0.3 mm in Figure 12. It can be calculated that the total critical current is 301 A/cm.

Location of Change: Page 9    line 275-283

  1. Reviewer2 wrote:

Conclusions

When the film thickness from 54 nm to 720nm, The RMS value rapidly rises from 1.6nm to 25nm"”

Please remove the capital “T” marked in red.

Our Response:

We greatly appreciate this question, which help us further improve paper quality. We totally agree with the suggestion and increase the recent advances in epi-MgO as follow:

When the film thickness from 54 nm to 720 nm, the RMS value rapidly rises from 1.6 nm to 25 nm and the in-plane texture increases from 5.4° to 2.2°.

Location of Change: Page 11    line 300-302

Reviewer 3 Report

Authors must adhere to the standards accepted in scientific journals when describing the deposition and analytical methods used in the work. Indicate the name of the installations used in the deposition of all studied materials. If these are original (homemade), then give a full description or a brief one but with reference to an earlier work. Editing of the layout is also necessary: start line 106 (RHEED) with a new paragraph. and in this place add a paragraph about oxygen addition (lines 118-122). In addition, there are repetitions in these paragraphs.
Regarding RHEED. It remains unclear what it means to characterize the thickness of the film using this method (line 106). Further (line 115) the words “not shown” sound as if the article contains a diagram of the deposition installation. And also, as if the QCM is installed on the crucible (measuring...the evaporation rate). Although usually the QCM is installed at the location of the substrate.
Regarding XRD. It is necessary to indicate which diffractometer was used. Lines 123-150. A fairly detailed description of the measurement geometry is given, which requires illustration. I think it's better to link to a well-cited review on the topic.

And further. Quite a stupid question from a non-expert. What was the substrate like in the “physical” sense? I mean: a plate of such and such a size, on which it is deposited....

And another question from a non-expert. I did not find in section 2 (Experiment) a description of the method of deposition of YBCO film (line 258).

Author Response

Dear Editor

Thank you so much for reviewing our work. We are delighted to be informed with a major review. The reviewers’ suggestions are very helpful to improve our manuscript. Here we submit our revised manuscript (the changes are highlighted in yellow) and point-to-point responses. We believe that we have improved the English writing.

Sincerely,

Yan Xue

Revisions in response to the Editors’ comments:

  1. Reviewer3 wrote:

Authors must adhere to the standards accepted in scientific journals when describing the deposition and analytical methods used in the work. Indicate the name of the installations used in the deposition of all studied materials. If these are original (homemade), then give a full description or a brief one but with reference to an earlier work. Editing of the layout is also necessary: start line 106 (RHEED) with a new paragraph. and in this place add a paragraph about oxygen addition (lines 118-122). In addition, there are repetitions in these paragraphs.

Regarding RHEED. It remains unclear what it means to characterize the thickness of the film using this method (line 106).

Further (line 115) the words “not shown” sound as if the article contains a diagram of the deposition installation. And also, as if the QCM is installed on the crucible (measuring...the evaporation rate). Although usually the QCM is installed at the location of the substrate.

Regarding XRD. It is necessary to indicate which diffractometer was used. Lines 123-150. A fairly detailed description of the measurement geometry is given, which requires illustration. I think it's better to link to a well-cited review on the topic.

Our Response:

We greatly appreciate this question, which help us further improve paper quality. We totally agree with the suggestion and increase the recent advances in IBAD-MgO as follow:

In this work, a home-made electron beam evaporation, which offers the ability to evaporate metal oxide materials with high melting point, was used to deposit epi-MgO films on 10 nm-thick IBAD-MgO films. In this experiment, the vacuum chamber was pumped down to 10-5 Pa using a molecular pump. Subsequently, the growth temperature (150 °C ~ 500 °C) required for the experiment was controlled by a home-made heating device. A reel-to-reel system enables the continuous preparation of long tapes for the dynamic deposition of the epi-MgO films. When the metal oxide films are prepared using electron beam evaporation, the ionic bonds of the materials can be easily disrupted by the electron beam, potentially causing the composition of the film material to deviate and affecting the microstructure of the film. Therefore, it is necessary to introduce oxygen to maintain the proper proportion of MgO film composition during the epi-MgO deposition process. The oxygen is introduced to the evaporation chamber through an oxygen valve. The quartz crystal microbalance (QCM) rate monitor and ion probe are used to measure the electron beam evaporation rate and the ion beam current density, respectively. The QCM is installed at the location of the substrate. Subsequently, the LMO cap layer is deposited on the epi-MgO by the DC reactive sputtering technology. Finally, a home-made metal organic chemical vapor deposition system is used to deposited YBCO films on the LMO/epi-MgO/IBAD-MgO/SDP-Y2O3 buffer layer. The detailed YBCO deposition process can be seen elsewhere [1,9].

The biaxial texture was characterized in situ using a high-energy electron diffraction (RHEED) equipment. The picture would exhibit regular diffraction spots array with biaxial textured MgO films. RHEED involves emitting a beam of high-energy electrons (5~100 keV) from an electron gun, which is incident on the sample surface at a small grazing angle (1~5°) to generate an electron diffraction beam. The crystal structure is then displayed by collecting the signal on a fluorescent screen. The RHEED comes in at a shallow incident angle which is perpendicular to the plane of the ion gun. The diffracted pattern is incident on a phosphorescent screen and the resulting image is then collected by a charge coupled device (CCD) camera, which is interfaced to a computer. The electron beam evaporator is placed off-center to allow the ion gun full angular range. The substrate is rotatable azimuthally about the substrate normal.

X-ray diffraction (XRD) (Bede D1) was used to probe the orientation and microstructure of MgO films. We used symmetric scan geometry for the majority of the XRD measurements. The term symmetric scan keeps the orientation of the scattering vector (q) fixed relative to the sample. In more practical terms, the inclined angle θ and are locked and changed in a ratio of 1:2, respectively. The magnitude of q varies as the angles are changed. For the θ-2θ scan geometry, the change in angle corresponds to a change in the lattice spacing and both the orientation and phase of a sample can be observed in the collected scan. The detailed description of the θ-2θ scan can be seen elsewhere [41].

Location of Change: Page 3    line 103-140

  1. Reviewer3 wrote:

And further. Quite a stupid question from a non-expert. What was the substrate like in the “physical” sense? I mean: a plate of such and such a size, on which it is deposited....
Our Response:

We greatly appreciate this question, which help us further improve paper quality.

The 20 m-length YBCO tapes deposited on LMO/epi-MgO/IBAD-MgO/SDP-Y2O3 template

In our library, we have fabricated YBCO tapes for transmission cables. The length and the width of the tape are 20 m and 10 mm, respectively.

  1. Reviewer3 wrote:
    And another question from a non-expert. I did not find in section 2 (Experiment) a description of the method of deposition of YBCO film (line 258).

Our Response:

We greatly appreciate this question, which help us further improve paper quality. We totally agree with the suggestion and increase the recent advances in IBAD-MgO as follow:

Finally, a home-made metal organic chemical vapor deposition system is used to deposited YBCO films on the LMO/epi-MgO/IBAD-MgO/SDP-Y2O3 buffer layer. The detailed YBCO deposition process can be seen elsewhere [1,9].

Location of Change: Page 3    line 118-121
